# Advances and Challenges of Stimuli-Responsive Nucleic Acids Delivery System in Gene Therapy

**DOI:** 10.3390/pharmaceutics15051450

**Published:** 2023-05-10

**Authors:** Meng Lin, Xianrong Qi

**Affiliations:** 1Beijing Key Laboratory of Molecular Pharmaceutics and New Drug Delivery System, School of Pharmaceutical Sciences, Peking University, 38 Xueyuan Road, Beijing 100191, China; 2Department of Pharmacy, West China Hospital, Sichuan University, Chengdu 610044, China

**Keywords:** gene therapy, stimuli-responsive, delivery system, nucleic acids

## Abstract

Gene therapy has emerged as a powerful tool to treat various diseases, such as cardiovascular diseases, neurological diseases, ocular diseases and cancer diseases. In 2018, the FDA approved Patisiran (the siRNA therapeutic) for treating amyloidosis. Compared with traditional drugs, gene therapy can directly correct the disease-related genes at the genetic level, which guarantees a sustained effect. However, nucleic acids are unstable in circulation and have short half-lives. They cannot pass through biological membranes due to their high molecular weight and massive negative charges. To facilitate the delivery of nucleic acids, it is crucial to develop a suitable delivery strategy. The rapid development of delivery systems has brought light to the gene delivery field, which can overcome multiple extracellular and intracellular barriers that prevent the efficient delivery of nucleic acids. Moreover, the emergence of stimuli-responsive delivery systems has made it possible to control the release of nucleic acids in an intelligent manner and to precisely guide the therapeutic nucleic acids to the target site. Considering the unique properties of stimuli-responsive delivery systems, various stimuli-responsive nanocarriers have been developed. For example, taking advantage of the physiological variations of a tumor (pH, redox and enzymes), various biostimuli- or endogenous stimuli-responsive delivery systems have been fabricated to control the gene delivery processes in an intelligent manner. In addition, other external stimuli, such as light, magnetic fields and ultrasound, have also been employed to construct stimuli-responsive nanocarriers. Nevertheless, most stimuli-responsive delivery systems are in the preclinical stage, and some critical issues remain to be solved for advancing the clinical translation of these nanocarriers, such as the unsatisfactory transfection efficiency, safety issues, complexity of manufacturing and off-target effects. The purpose of this review is to elaborate the principles of stimuli-responsive nanocarriers and to emphasize the most influential advances of stimuli-responsive gene delivery systems. Current challenges of their clinical translation and corresponding solutions will also be highlighted, which will accelerate the translation of stimuli-responsive nanocarriers and advance the development of gene therapy.

## 1. Introduction

Gene therapy, a powerful therapeutic strategy for cancer, represents a breakthrough in managing diseases that were previously considered incurable. To date, cancer is by far the most common disease treated by gene therapy. It comprises over 60% of all ongoing clinical gene therapy trials worldwide, followed by monogenetic and cardiovascular diseases [1]. Gene therapy works by introducing exogenous nucleic acids to alter the disease-related gene, such as gene deletion, gene knockin, gene replacement and gene suppression. Typically, nucleic acids for therapeutic purposes can be classified into antisense oligonucleotides (ASOs), small interfering RNA (siRNAs), microRNA (miRNA), mRNA, plasmids, polypurine reverse hoogsteen (PPRH) hairpins and clustered regularly interspaced short palindromic repeats (CRISPRs)-based machinery, which differ in size and mechanisms of action [2]. ASOs are single-stranded oligonucleotide molecules, usually containing 15–25 nucleotides. They can downregulate the target gene expression through the induction of RNaseH-dependent RNA degradation or modulate the splicing processes by binding to a splicing regulatory region [3]. In general, siRNA and miRNA are employed to reduce the gene expression by triggering the mRNA degradation or affecting the stability of mRNA [4]. On the contrary, mRNA and plasmids usually serve as the template to enhance the gene expression to offset the loss of function of the disease-related genes. PPRHs are DNA hairpins formed by intramolecular reverse Hoogsteen bonds, and they can bind to double-stranded DNA by Watson:Crick bonds, which results in the displacement of the polypurine strand and the specific inhibition of gene transcription [5]. Despite these functional differences, as nucleic acids, they share common characteristics. For example, they are all hydrophilic and negatively charged macromolecules, and the presence of various biological membranes (cellular membrane, nuclear membrane and organelle membrane) severely hinder their uptake. Unlike small molecules, which can easily pass through the biological membranes via passive diffusion, macromolecules are less likely to cross biological membranes due to their high molecular weight [6]. Furthermore, the massive negative charges of nucleic acids also prevent their cellular uptake due to electrostatic repulsion. Moreover, the half-lives of nucleic acids are very short, as they are degraded by enzymes during circulation [2,7]. The harsh environment in vivo, such as acidic endosome and lysosome, may also destroy nucleic acids [8]. Considering the aforementioned dilemma of nucleic acids for therapeutic use, such as instability and membrane impermeability, it is urgent to develop efficient methods to overcome the disadvantages of nucleic acids to broaden their application field. To date, physical methods, viral vectors and non-viral vehicles have been applied to facilitate the delivery of nuclear acids [9,10,11]. However, the potential damage to the cells limits the therapeutic use of physical methods in vivo [11,12]. Likewise, despite the high delivery efficiency, the risks of genomic integration and the immunogenicity of utilizing viral vectors remain the main concerns for their clinical use [13,14]. On the contrary, nanoparticles have become alternatives of viral vectors owing to their extraordinary properties. Generally, nanoparticles are composed of biocompatible materials, which guarantees their safety and avoids the risks of genomic integration. Further, nanoparticles can be produced at a large scale and are easy to engineer, which is beneficial for their clinical translation and surface modification. For efficient gene delivery, the nanoparticles must perform multiple functions to assist the nucleic acids in overcoming the various extracellular and intracellular barriers. They have to maintain stability during circulation, evade clearance by the mononuclear phagocytic system (MPS) and accumulate preferentially at the disease sites after extravasating from the vessels. The most challenging aspect of the intracellular barriers is that the nanoparticles must be capable of protecting the loaded nucleic acids from degradation of the lysosome and timely releasing the nucleic acids to their target sites (cytoplasm or nucleus) [15,16].

By far, numerous nanoparticles have been utilized to deliver nucleic acids for gene therapy. As the properties of nanoparticles (size, zeta potential, stability and release profile) play a central role in determining their fate in vivo [17,18], these parameters must be taken into consideration when designing a gene delivery system. Among these nanoparticles for gene delivery, polymer-based polyplexes account for a large part of non-viral vectors. These positively charged polymers, usually containing ionizable amine (N) groups, can complex with the nucleic acids containing phosphate (P) groups via electrostatic interactions and form uniform polyplex at the appropriate N/P ratio, which represents a critical indicator to evaluate the gene condensation capacity. However, the positively charged polyplexes are prone to protein adsorption and form “protein corona”, which has a determining effect on the in vivo fate of nanoparticles [19]. Generally, most of the proteins adsorbed on the nanoparticle surface speed up the clearance of nanoparticles [19], and these proteins are termed as opsonins, which could facilitate the recognition and phagocytosis by antigen-presenting cells (APCs), such as macrophages and dendritic cells (DCs), to exhibit opsonization [20]. Moreover, the size of nanoparticles increases upon protein absorption [21,22], making it harder for them to extravasate from the vessels and penetrate the deep tumor. Therefore, researchers attempt to reduce the protein adsorption of positively charged nanoparticles by shielding their positive charges using a negatively charged coating, such as biocompatible ionic polymers and extracted cell membranes [23,24,25,26]. These strategies do prolong the circulation time of nanoparticles to some extent, but also diminish cellular uptake, as the nanoparticles of negative charges are less likely to pass through the negatively charged cell membranes. Therefore, taking advantage of the pH difference between normal tissues and tumors, pH-responsive nanoparticles were fabricated to solve the problem [27,28]. These nanoparticles can be protected by their coating during circulation and selectively take off the coating for efficient cellular uptake in response to the acidity in the tumor microenvironment. These stimuli-responsive nanoparticles provide the possibility of controlling drug release in a spatiotemporally controlled manner and improving the specificity of gene therapy. Inspired by the pathological changes (pH, redox and enzymes) of tumors, researchers have devoted great efforts to developing stimuli-responsive nanoparticles, as well [29,30]. The rapid development of nanomedicine has also flourished in the field of smart nanoparticles. In addition to the endogenous stimuli, some exogenous stimuli have been harnessed to regulate the delivery process of nanoparticles [31], largely relying on the unique properties of magnetic nanoparticles, light-sensitive nanoparticles and ultrasound-responsive nanoparticles, etc. Under some circumstances, several stimuli are combined to more precisely regulate the in vivo site of nanoparticles.

This review provides an overview of the nanoparticles used for cancer gene therapy and places emphasis on the stimuli-responsive nanoparticles. We will briefly introduce the different kinds of nucleic acids for therapeutic purposes and the currently delivery methods for nucleic acids, with an emphasis on the multiple barriers for in vivo gene delivery. In addition, some design criteria concerning effective delivery efficiency will be mentioned, as well. Moreover, we will elaborate the principles of stimuli-responsive nanocarriers and their recent advances in tumor treatment over the last few years, and discuss the current challenges of gene delivery using stimuli-responsive nanocarriers in the clinic.

## 2. Classification of Nucleic Acids

Therapeutic agents for gene therapy contain varieties of nucleic acids that differ in size, stability, intracellular transport and action mode [32]. These differences will have a huge impact on the encapsulation efficiency and the delivery efficiency, and thus the characteristics of the specific nucleic acids should be taken into consideration when designing a delivery system.

siRNA and miRNA are the smallest nucleic acids, usually consisting of about 20 base pairs (bps). SiRNA are short, duplex, non-coding RNA, and they can associate with various nucleases in the body to form RNA-induced silencing complex (RISC), which can specifically induce the degradation of mRNA to silence the expression of target genes upon binding with their complementary target mRNA [33]. Similarly, miRNAs are a class of non-coding RNA molecules and can mediate mRNA translational inhibition or degradation by binding to complementary target mRNAs [4], which play a central role in cell differentiation, proliferation and survival [4,34]. In addition to these small RNAs, RNAs of larger size, such as message RNAs (mRNAs), have shown potential in gene therapy. Generally, mRNA molecules are composed of 1000–15,000 bases with a molecular weight of 300–5000 kDa [35], nearly a hundredfold of siRNA and miRNAs in length, which makes it even more challenging to deliver mRNAs for therapeutic use. Unlike siRNA or miRNAs that inhibit gene expression, mRNAs can direct the protein synthesis based on their carried gene information to enhance gene expression, which is quite beneficial to treating diseases caused by protein misexpression or disfunction.

DNA molecules are the largest among these nucleic acids, ranging from several to hundreds of kilobases (kb). Plasmids, the most commonly used DNA molecules in gene therapy, possess a closed, circular, double-stranded structure and are capable of replicating autonomously. By genetically engineering the plasmids, researchers can insert specific gene segments into the plasmids and initiate the gene expression in cells by virtue of the plasmids’ replication ability. Therefore, plasmid-based gene therapies are suitable for the treatment of genetic diseases that are currently considered incurable. Additionally, they have also shown potential in cancer treatment.

Clustered regularly interspaced short palindromic repeats (CRISPRs)/CRISPR-associated proteins (Cas), the recently developed genetic editing tools, have revolutionized the field of gene editing [36,37]. CRISPR/Cas9, the most extensively studied form of genetic editing machinery, contains a Cas9 enzyme and a single guide RNA (sgRNA), which can cleave the target gene at the desired site upon binding with the target gene via complementary base pairing. The resulting double-strand breaks (DSBs) can initiate the repair mechanisms of cells—homologous end joining (NHEJ) or homology-directed repair (HDR)—leading to gene knockout or gene knockin [38,39]. In cancer treatment, NHEJ-mediated gene knockout has been widely utilized to inhibit the expression of genes associated with tumor progression, such as Plk1, MTH1, HPV E7 and PTPN2 [40,41,42,43,44]. The key to CRISPR-based genetic editing lies in the selection of sgRNA, as the sgRNA directs the Cas9 to the target site for cleavage. In practice, the CRISPR/Cas9 machinery contains three formats—plasmid, mRNA and Cas9/sgRNA ribonucleoproteins (RNPs)—and they vary in size, zeta potential, stability and immunogenicity. Notably, the length of plasmid encoding Cas9 and sgRNA is nearly 10 kb, much larger than the ordinary plasmid, making it an efficient delivery system for ordinary plasmid less efficient in delivering the CRISPR/Cas9 plasmid.

After briefly introducing the different types of nucleic acids for therapeutic use, it is essential to point out their limitations in use. Firstly, the naked nucleic acids are unstable and prone to degradation by nucleases in circulation, resulting in a relatively short half-life period [2,7]. Secondly, nucleic acids are impermeable to the cellular membranes due to their large molecular weight and massive negative charges, leading to low cellular uptake and, thus, low gene transfection efficiency, which is one of the most vital parameters to determine gene delivery efficiency [2]. Thirdly, their potential off-target effect may cause damage to normal tissues. To overcome these limitations of nucleic acids, researchers have tried diverse methods, for example, modifying siRNA at the desired site including the 2′-position, phosphate linkage and ribose to improve its nuclease resistance in serum [45]; conjugating siRNA to diverse ligands, including peptides, antibodies, aptamers and small molecules, to enhance cellular uptake and targetability [46,47,48]. Although these strategies offer an opportunity to overcome the shortcomings of siRNA, they are not widely used due to the complexity of manipulation. Alternatively, efficient delivery strategies may better advance the application of nucleic acids.

In general, gene delivery strategies can be classified into physical, viral and non-viral approaches. The physical approaches include electroporation, microinjection, induced transduction by osmocytosis and propanebetaine (iTOP), mechanical cell deformation and hydrodynamic injection, and most of them are only suitable for in vitro use. Electroporation, an extensively used approach to deliver nucleic acids into cells, facilitates nucleic acid entering cells by promoting the permeability of the cell membrane. Despite its high transfection efficiency, it may cause cell death and is restricted to in vitro use. Hydrodynamic injection works by rapidly administrating a large volume (equivalent to 8–10% of the body weight) of a nucleic acid solution, which facilitates nucleic acids to enter the cells by virtue of hydrodynamic pressure-induced formation of pores on the cell membranes [12]. Although it is by far the simplest way for in vivo gene delivery in small animals, the adverse effects related to hydrodynamic injection, such as elevated blood pressure, liver expansion, cardiac dysfunction and even animal death, hinder their clinical use, limiting their therapeutic use in vivo [11]. On the contrary, more and more efforts have been devoted to developing viral and non-viral delivery approaches. Basically, viral approaches refer to utilizing viral vectors to carry the exogenous gene into cells for gene expression. To date, the majority of the therapeutic gene agents approved for clinical use rely on viral vectors. Adenovirus, adeno-associated virus (AAV) and lentivirus are the most commonly used viral vectors. Gendicine™, which was approved for clinical use in 2013, is an adenoviral vector [1]. Taking the most promising viral vectors—AAV, for example—they contain a single-stranded DNA genome of about 4.7 kb and have more than 200 variants [49]. Moreover, they are capable of transfecting dividing and non-dividing cells with high transfection efficiency and non-pathogenicity, demonstrating great potential in gene delivery. Although AAV overcomes the potential risks of lentivirus that integrates into the host genome, the packaging limit (approximately 4.5 kb for AAV) [50] and concerns about its immunogenicity remain the critical issues to be addressed. By contrast, the emergence of numerous nanoparticles has brought light to the gene delivery field and aroused the great interest of researchers. Compared with viral vectors, nanoparticles have the advantages of facile preparation, good biocompatibility and stability, and low cost. More importantly, they are unrestrained by the packaging size and, thus, are capable of delivering genes of different sizes, even the large plasmid that encodes the CRISPR/Cas9 machinery for genetic editing. In addition, it has been well accepted that nanoparticles (≤200 nm) are prone to accumulating in the tumors via the enhanced permeability and retention (EPR) effect, which becomes the rationale of using nanoparticles for tumor targeting. By modifying the nanoparticles with ligands, researchers can further increase the targetability of nanoparticles via the ligand–receptor interactions, and this strategy has been widely adopted in the nanomedicine field.

## 3. Multiple Barriers of Gene Delivery In Vivo

As we mentioned above, naked nucleic acids are susceptible to nucleases and impermeable to cell membranes. Therefore, polymers of positive charge are utilized to condense the negatively charged nucleic acids to form uniform nanoparticles via electrostatic interactions, which can effectively protect the nucleic acids from nucleases to improve their stability and facilitate their cellular uptake. In addition to the package issues, nanoparticles encounter multiple barriers in vivo, as discussed below. Firstly, nanoparticles are in a complicated environment that contains various chemical substances, such as proteins and nucleases, during circulation, which are likely to have an adverse effect on the nanoparticles. For example, serum proteins are prone to adhering to the surface of nanoparticles to form the “protein corona”, and the presence of “protein corona” may significantly change the characteristics of the nanoparticles, such as the zeta potential, stability and biodistribution [51,52,53]. Additionally, nanoparticles must be able to avoid clearance by the MPS to maintain a relatively long circulation time. As the size and zeta potential are key parameters to determine the clearance of nanoparticles, the surface properties of the nanoparticles should be taken into consideration. Generally, nanoparticles smaller than 10 nm tend to be filtrated by the kidney, while those larger than 100 nm are more likely to be captured by the MPS. The positively charged nanoparticles are also more prone to clearance than their positively charged counterparts. PEGylation, one kind of modification strategy that conjugates the nanoparticles with polyethylene glycol (PEG), has been extensively adopted to prolong the circulation time of nanoparticles [26]. Some biomimetic strategies, such as coating nanoparticles with cell membranes, have also been introduced in nanomedicine recently to improve the circulation stability of nanoparticles [23,24,25]. Secondly, ideal vesicles should have the ability to accumulate to their target organs. Therefore, they must cross the barriers formed by the vascular endothelial cells and extravasate from the vessels. Additionally, the massive extracellular matrixes in tumors also makes it challenging to achieve deeper penetration. Thirdly, the nanoparticles should target specific cells more than tissues and overcome multiple intracellular obstacles, as there are varieties of cells, including tumor cells, vascular endothelial cells and various immune cells, in the tumor microenvironment. After entering their target cells, the encapsulated nucleic acids may be damaged by the enzymes or acidic environment of the endosome/lysosome. Therefore, efficient endosomal escape is vital for the nucleic acids to maintain their biological activities [54,55]. PEI, the most classic cationic polymer, is capable of rapidly escaping from endosome via the “proton sponge effect” and has been introduced into the vesicles either alone or for modification [44,56,57]. Other strategies to evade the endosome–lysosome pathway have also been attempted recently. In a study, Qiu et al. decorated nanoparticles with the endoplasmic reticulum (ER) membrane derived from cancer cells to change the intracellular transport behavior of these nanoparticles [58]. Unlike most cationic vesicles that enter the cells via clathrin-mediated endocytosis and subsequently are trapped in the lysosome, the ER-membrane-decorated nanoparticles utilized the endosome–Golgi apparatus/ER pathway and evaded lysosomal degradation. Apart from the aforementioned endosome escape, the encapsulated nucleic acids must be released on time and reach their target sites to fulfill their functions, which may vary with the types of nucleic acids. For siRNA, miRNA and mRNA that fulfill their biological functions in the cytoplasm, they just need to be released in the cytoplasm. Yet, for plasmids whose target sites are located in the nucleus, they have to overcome the barriers formed by the nuclear envelope before entering the nucleus. As the macromolecules cannot easily diffuse into the nucleopore, the disassembly of the nuclear envelope may account for the nuclear localization of plasmids during cellular mitosis [2]. For non-dividing cells, the plasmids transport into the nucleus via the nuclear pore complex, which is critical to regulating the transport of various macromolecules.

In summary, to successfully deliver nucleic acids into the target cells, nanoparticles must have the following characteristics (Figure 1): (1) efficiently condense the nucleic acids, (2) remain stable in circulation and evade the clearance of MPS, (3) accumulate at the target organ, (4) overcome the extracellular obstacles for deep penetration, (5) achieve efficient cellular uptake in target cells, (6) evade the lysosomal degradation and release the nucleic acids on time, (7) facilitate the nuclear location of plasmids. Additionally, excellent nanoparticles should also be biocompatible and biodegradable from the perspective of safety. Moreover, site-specific and time-controlled release delivery systems are demanding for therapeutic use, especially in the treatment of diseases such as cancer, which will be discussed in the following section.

## 4. Stimulus-Responsive Nanocarriers

For the past few decades, a variety of nanocarriers have been employed in the gene delivery field, which encompasses various organic nanoparticles, inorganic nanoparticles and biomimetic nanoparticles, such as liposomes [59], polymeric nanoparticles [60,61,62,63], gold-based nanoparticles [64,65,66], calcium carbonate nanoparticles [67,68], metal organic frameworks (MOFs) [69,70] and exosomes [71,72,73]. As the most typical gene delivery systems, PEI-based polymeric nanoparticles and cationic lipids-based liposomes have been commercialized and extensively employed to transfect various cells. However, unsatisfactory transfection efficiency and cytotoxicity remain the obstacles for their in vivo use. To solve these problems, researchers have made tremendous efforts to engineer the polymers, and readers may refer to the review by Yang et al. that discussed in detail the different strategies to modify the polymers [55]. In this section, we will focus on the recent advances of stimulus-responsive nanocarriers. Generally speaking, stimulus-responsive nanocarriers can respond to external or internal stimuli to more precisely and smartly control nanocarriers’ in vivo behaviors. Considering the differences between tumor and normal tissues in pH, redox condition and expression of enzymes, these internal signals are of great use in fabricating nanocarriers that are capable of regulating the gene release in a tumor-specific manner. Moreover, some external stimuli, including light, magnetic field and ultrasound, have also been utilized to spatiotemporally control the transport of nanocarriers (Figure 2).

### 4.1. pH-Responsive Nanocarriers

The pH is highly heterogeneous and varies with the physiological environment. For example, the extracellular environment of tumors tends to be more acidic (pH ≈ 6.5) than that in normal tissues (pH 7.4), and the pH of endosome/lysosome is even lowered to 5. Therefore, the heterogeneity of pH has been broadly harnessed to enhance cellular uptake and control drug release, etc.

#### 4.1.1. pH-Responsive Detachment of Outlayer Coating

PEGylation is a common strategy used in gene delivery system to prolong the circulation time. However, coating of PEG shields the positive charges of the polyethylenimine/DNA complex and may limit the cellular uptake, endosomal escape and subsequent transfection efficiency. To overcome the dilemma of PEG coating, Guan et al. designed a pH-responsive, detachable, polyethyleneglycol (PEG) shielding gene delivery system [27]. The aldehyde-modified mPEG was coated onto PEI/DNA via pH-responsive Schiff base bonds. The PEG-shielded nanoparticles were stable in normal tissue during the transport process, while at the tumor site, the PEG was detached due to the cleavage of Schiff base bonds by the slightly acidic tumor extracellular pH. The re-exposed PEI/DNA could effectively cross the cellular membranes and result in gene expression. Similarly, using a pH-sensitive imine bond to conjugate the PEG derivative O’-methylpolyethyleneglycol (omPEG) into lipid, a PEG-detachable nanoparticle was designed for the tumor-directed delivery of chemo- and gene therapies [74]. The nanoparticles were coated with multifunctional peptides and a PEG derivative to decrease noncancerous cellular uptake, while in the acidic tumor environment, the detachment of PEG rendered the exposure of peptides for active tumor targeting via specific ligand–receptor binding. The pH-detachable nanoparticles demonstrated improved cellular uptake in HNC human tongue squamous carcinoma (SAS) cells, and the released miR-200 and irinotecan synergistically improved the therapeutic efficacy in a SAS tumor-bearing mouse model.

Apart from pH-sensitive PEG detachment, pH-responsive outlayer detachment has also been reported [75]. In this study, miR146a was conjugated to the polyethylenimine-(4-(bromomethyl) phenylboronic (PEI-PBA) via the ester linkage, and the formed inner core was further shielded by an outer-layer polyplex PEI−DMA-C225 through the electrostatic interactions. The outer-layer polyplex of nanoparticles exposed positively charged amino groups again in the endo/lysosomes, and thus detached from the nanoparticles. Afterwards, the intracellular ATP triggered the release of miR146a by competitively binding with the PEI-PBA. This pH/ATP-activated complex significantly suppressed invasion, colony formation and migration in DU145 cells and inhibited tumor growth in vivo, which demonstrated the potential in the treatment of androgen-independent prostate cancer.

#### 4.1.2. pH-Responsive Gene Release

Various nanomaterials, including the metal–organic framework [69], micelle [76], CaP-phospholipid complex [77], calcium carbonate nanoparticles [78], quantum dot [79], black phosphorus nanosheet [80] and polymer [81], have been used for pH-triggered gene release. For example, taking advantage of the pH sensitivity of CaP, an As_2_O_3_ and HER2-siRNA coloaded CaP-phospholipid complex nanocarrier (AH RNP) was constructed to control drug release in an acidic lysosome environment [77]. Unlike the free Cy5siRNA, which showed a burst release profile, regardless of the pH, the Cy5siRNA in AH RNPs demonstrated a pH-dependent release profile: its release rate was faster at pH 5.5 than that at pH 7.4 due to the dissociation of CaP in the acidic environment. Moreover, the degradation of nanomaterials could facilitate their clearance from the body to minimize the toxicity [82,83].

Lipid Nanoparticles (LNPs) have been widely used in the gene delivery field, and some LNP-based drugs have been approved (e.g., Onpattro for the treatment of amyloidosis) [9,84]. LNPs are composed of ionizable lipid, PEGylated lipid, phospholipid and cholesterol. As a critical component of LNP, ionizable lipid plays a key role in LNP potency [9]. Unlike permanently charged cationic lipids, ionizable lipids are uncharged at the physiological pH and become positively charged at an acidic pH. This feature renders the LNP the characteristic of pH-responsive gene release [85]. In the acidic environment of an endosome, the ionizable lipids become protonated and interact with the negatively charged endosomal membrane, which leads to the destabilization of the endosomal membrane and facilitates the endosomal escape of nucleic acids [86,87].

Considering the heterogenicity of different tumors, multimodal therapeutic strategies should be adopted to achieve effective anti-tumor efficacy. Black phosphorus (BP), a dimensional nanomaterial with a high surface-to-volume ratio, has attracted interest in biomedical research recently. Using PEG and PEI dual-functionalized BP nanosheets (PPBP), Chen et al. constructed a degradable gene delivery system for cancer-targeted synergistic therapy [80]. The PPBP demonstrated high drug loading of human telomerase reverse transcriptase (hTERT) siRNA and exhibited potent photodynamic therapy/photothermal therapy (PDT/PTT) activities. In a low pH and reactive oxygen species (ROS)-rich environment, the PPBP gradually degraded and endowed the release of hTERT siRNA to fulfill its biological functions. Combination of PTT, PDT and siRNA-mediated hTERT knockdown synergistically suppressed tumor growth and metastasis in the mouse model.

#### 4.1.3. pH-Responsive Charge Reversal

Tissue factor (TF), a critical initiator of blood coagulation, is upregulated in a variety of tumors and plays a vital role in tumor progression and metastasis [28]. Targeting tumor-associated TF has shown efficacy in anticancer metastasis and the prevention of cancer-associated abnormal hypercoagulability. To overcome the side effects of TF silence in normal tissues, such as tissue bleeding complications, strategies to specifically deplete TF in tumors are highly desired. In this context, Liu et al. developed a peptide-based gene delivery system that specifically reduced TF expression in tumors [28]. An amphipathic peptide was used as the scaffold to complex with TF siRNA, and to improve the delivery efficiency, a cyclic RGD (cRGD) peptide that shows high affinity to the overexpressed αvβ3 on most tumors and a polyhistidine sequence that is responsive to the slightly acidic microenvironment in tumor tissues were introduced. The pH-sensitive sPNPs were sensitive to the acidity in the tumor microenvironment and reverted their potentials from negatively charged to positively charged. In combination with the recognition of cRGD-αvβ3, the sPNPs efficiently facilitated cellular uptake. In the lysosomes, the sPNPs were depolymerized and released siRNA by virtue of the protonation effect of polyhistidine. In a 4T1-breast-tumor-bearing mouse model, sPNPs suppressed the expression of tumor-associated TF by approximately 75% and significantly inhibited lung metastasis, outperforming pH-insensitive PNPs.

### 4.2. ROS-Responsive Nanocarriers

Many diseases, including tumor, are characterized by a high level of reactive oxygen species (ROS). The reprogramming of redox metabolism induces abnormal accumulation of ROS in cancer cells [88], making ROS levels much higher in cancer cells (up to 100 μM) than in normal tissue (≈0.02 μM) [89,90]. In turn, to maintain the balance between the generation and depletion of ROS, tumor cells change antioxidant defense systems by upregulating the reducing substances. For example, glutathione (GSH), the main reducing substance in the tumor cytosol, maintains a concentration of about 2–10 mM, 100–1000 times higher than that in normal tissues [91]. Therefore, taking advantage of the differences in GSH expression, various redox-responsive gene delivery systems have been fabricated for targeted tumor therapy [82,92,93,94,95]. Among these designs, the most common strategy is to introduce a disulfide bond to endow the nanocarriers with the redox-sensitive characteristics, as the disulfide bond is prone to leakage in the presence of reducing substances, such as GSH. For instance, Nie et al. designed an unlockable core–shell nanocomplex (Hep@PGEA) that is composed of disulfide-bridged heparin nanoparticle (HepNP) core and low-toxicity PGEA cationic shell [82]. The nanocomplex could be specifically triggered by the reducing agents in cells and accelerated the release of MiR-499. This nanocomplex has also been utilized to load the pCas9 plasmid encoding Cas9 protein and sgRNA targeting oncogene survivin [96], and demonstrated potent anti-tumor efficiency by inhibiting proliferation, migration and invasion, as well as promoting apoptosis of hepatocellular carcinoma cells. In addition to the inclusion of a single disulfide bond, polydisulfide-based siRNA nanocomplexes have been developed [95]. Poly(disulfide amine) was synthetized via Michael addition and served as the backbone for GSH-sensitive breakdown, and further introduction of an arginine-rich poly((N,N′-bis(acryloyl)cystamine-coethylenediamine)-g-Nω-p-tosyl-L-arginine) (PBR) rendered the polycations to condense the siRNA. Due to the excellent GSH-responsiveness and transfection efficiency, the nanocomplexes significantly suppressed the proliferation and migration of tumor cells, with negligible cytotoxicity. Apart from polymers, biocompatible materials, such as DNA, have also demonstrated potential in gene delivery. In a recent study, Wang et al. designed a novel DNA nanostructure-based siRNA carrier with the DNA origami technique [93]. After the intercalation of DOX, siBcl2 and siP-gp (siRNA targeting Bcl2 and P-glycoprotein) were loaded in the inner space of the open nanodevice via disulfide bond-conjugated capture strands. Upon reduction by GSH in the cytoplasm, siRNAs were released from the tubular DNA nanodevice to silence the expression of Bcl2 and P-glycoprotein. In combination with DOX-induced cytotoxicity, this nanodevice elicited potent antitumor responses in vivo.

In addition to disulfide bonds, boric acid ester linkage has been included in fabricating redox-responsive nanomaterials. In the pioneering work reported by Shen, they devised a boric acid ester-based, charge-switchable, cationic polymer B-PDEAEA, whose phenylboronic acid group was prone to oxidation by the elevated intracellular ROS levels and converted to being negatively charged for ROS-controlled DNA release. Considering the unsatisfactory transfection efficiency of B-PDEAEA in the presence of serum, they further coated B-PDEAEA/DNA with a PEGylated fusogenic lipid to improve its stability, but the delivery efficiency was slightly enhanced [97]. Therefore, they replaced the fusogenic lipid with cationic lipids and developed a lipopolyplex-based gene carrier in a subsequent study [98]. The lipid coating improved the stability of the lipopolyplex, enhanced cellular uptake and promoted nuclear localization. Moreover, the ROS generated by the cationic liposome could in turn accelerate the charge reversal of the B-PDEAEA polymer and release DNA. Inspired by their work, some other boric acid ester-based gene delivery systems have also been reported [89,99].

Thioketal, the ROS-cleavable linker, has demonstrated potential in devising a ROS-responsive carrier for treating diseases, including intestinal inflammation and cancer [100,101]. For instance, Zheng et al. designed a photosensitizer-based gene delivery system by polymerizing the photosensitive drugs porphyrins via a thioketal linker [100]. The cationic polyporphyrins served as both the HIF-1α siRNA carrier and the photodynamic therapy agent due to their excellent photocytotoxicity. Under irradiation, polyporphyrins generated ^1^O_2_ to kill the cancer cells, and the high levels of intracellular ROS resulted in the degradation of polyporphyrins and the release of siRNA to knockdown HIF-1α, which cooperatively inhibited tumor growth by the combination of photodynamic therapy and gene therapy in a H22 xenograft tumors-bearing mice model.

### 4.3. Enzyme-Responsive Nanocarriers

Enzymes are bioactive substances that are highly specific and efficient in catalyzing their substrates, which play an essential role in various biological processes. Along with the tumor progression, the profiles of enzymes in tumors have also evolved. Taking advantage of the differences in enzyme expression between normal tissues and tumors, researchers have fabricated enzyme-sensitive nanocarriers for tumor-targeted gene therapy, including enzyme-triggered gene release, enzyme-controlled carrier dissociation, enzyme-enhanced cellular uptake, etc.

Esterases are a class of enzyme with the ability to hydrolyze ester bonds. Qiu et al. synthesized an esterase-responsive cationic polymer, PQDEA, which contains an acetyloxybenzyl ester [102]. This acetyloxybenzyl ester could be cleaved by the high intracellular esterase, leading to the elimination of p-hydroxymethylphenol and thus the charge reversal of the cationic polymer. Owing to the weaker interactions between the anionic polymer and the plasmid, the plasmid was released for TRAIL suicide gene expression. The LPQDEA demonstrated higher transfection efficiency than PEI 25K and lipo, and effectively promoted the apoptosis of tumor cells in the HeLa tumor model.

Matrix metalloproteinases are overexpressed in many tumors and are associated with tumor progression, invasion and metastasis [103]. Researches have indicated that matrix metalloproteinase-9 (MMP9), a type-IV collagenase/gelatinase, is overexpressed across various tumors, including colon, breast and gastric tumors [104,105,106]. Considering the high levels of MMP9 in the tumor tissues, Boehnke et al. constructed MMP9-sensitive nanotheranostics using layer-by-layer (LPL) technology. The liposome acted as an inner core, and poly-l-arginine (PLR), siRNA, and propargyl-modified polyl-aspartate (pPLD) were subsequently adsorbed onto it to improve colloidal stability and enhance the gene delivery efficiency. The MMP9-sensitive biosensor peptides were further conjugated to the LPL-coated liposome using copper(I)-catalyzed click conjugation chemistry. In the tumor microenvironment, biosensor peptides could be specifically cleaved and release the reporter fragments, which provided valuable information about the disease progression. In three mouse models including pancreatic, colorectal and ovarian cancer, the biosensor LbL nanoparticles demonstrated potent diagnostic capabilities and higher than 50% knockdown efficiency [107].

Cell-penetrating peptide (CPP) is a short peptide, usually composed of no more than 30 amino acids [108,109]. Due to its outstanding transduction ability and biocompatibility, CPP has been commonly utilized in the nanomedicine field [110]. However, the CPP-mediated cellular penetration is non-specific and may lead to undesired side effects for normal tissues. To address this issue, tumor microenvironment-sensitive polypeptides (TMSP) have been developed to overcome the shortcomings of traditional CPP. Generally, TMSP is composed of three segments: the cell-penetrating peptides (CPP, oligoarginine), the shielding peptides (EGGEGGEGGEGG) and a matrix metalloproteinases-2/9 (MMP-2/9)-cleavable peptide linker (PVGLIG, etc.) [111,112]. In the normal physiological environment, the CPP is shielded by negatively charged peptides and remains inert. While in the tumor microenvironment, the MMP-2/9-cleavable peptide linker is cleaved and re-exposes the CPP to enhance cellular uptake. Modification of TMSP onto nanoparticles has demonstrated great potential in promoting the cellular uptake of therapeutic agents, such as chemotherapeutics drug, siRNA and even plasmid [111,112,113,114,115]. For instance, we designed an amphiphilic dendrimer-engineered nanocarrier system (ADENS) for co-delivering paclitaxel and siRNA [114]. Due to its hollow core/shell structure, siRNA was encapsulated in the hydrophilic cavity, while paclitaxel was loaded in the hydrophobic interlayer. Compared with the unmodified ADENS, the TMSP-modified ADENS showed enhanced cellular uptake, tumor penetration and accumulation via an MMP-2/9-dependent mechanism. Moreover, we utilized the TMSP grafting strategy to construct a TMSP-responsive gene delivery system for CRISPR-based gene therapy [115]. The fluorinated polyethylenimine (PF) was chosen to condense the CRISPR plasmid owing to its superiority in improving cellular uptake, the endosomal escape of polymer/DNA polyplexes and intracellular DNA disassociation from the polymer [55,116,117]. A haluronic acid (HA)-PEG-TMSP (HPT) module was further introduced via electrostatic interactions (Figure 3). In the cellular uptake experiments, pretreating the cells with MMP inhibitor (GM6001) inhibited the cellular uptake of HPT-PFs, while adding MMP further increased the cellular uptake. These results suggested the modification of TMSP indeed promoted the cellular uptake. Moreover, we found that HPT-modified nanoparticles demonstrated significantly higher transfection efficiency than unmodified nanoparticles, suggesting the dual modification of HA and TMSP-facilitated gene delivery. In addition to TMSP-mediated cellular uptake, MMP-responsive PEG detachment has also been reported for efficient gene therapy [118].

Hyaluronidases, the enzymes to degrade HA, are expressed in many tissues [119]. Most members of the hyaluronidase family participate in tumor progression [120]. Studies have indicated the high level of hyaluronidase in some tumors, such as breast and prostate cancer [121,122]. Therefore, hyaluronidases could be used to trigger HA degradation for gene release [123,124,125,126]. Choi et al. developed a versatile RNA interference nanoplatform capable of delivering both siRNA and miRNA [127]. This HA-based nanoplatform can actively target tumor cells by virtue of the high affinity of HA to CD44 receptors (usually highly expressed on tumor cells), and after the CD44 receptor-mediated endocytosis, its CaP layer dissolved in the acid lysosome, and the nanoplatform was degraded by intracellular hyaluronidase, leading to the release of all payloads. Apart from the hyaluronidase-triggered degradation of a HA-based nanocarrier, a novel hyaluronidase-triggered charge reversal polymetformin (PMet)-based nanocarrier has also been devised for co-delivering doxorubicin (DOX) and plasmid encoding the IL-12 gene (pIL-12) to treat metastatic breast cancer [125]. PMet-based polycationic micelles incorporated DOX in the inner part and absorbed pIL-12 on the surface. To minimize the interference of proteins in blood and improve the nanocarrier’s stability, the polycationic micelles were further coated by thiolated hyaluronic acid (HA-SH) via electrostatic interaction and a thiol crosslink. In the presence of hyaluronidase, HA-SH was deshielded, which rendered the exposure of cationic PMet. The subsequent protonation of PMet in endo/lysosome conditions promoted the release of pIL-12 and DOX. This nanocarrier exhibited superior antitumor efficacy and anti-metastasis efficacy in in the 4T1 tumor model mice.

### 4.4. ATP-Responsive Nanocarriers

Adenosine triphosphate (ATP) is one of the most abundant molecules in the cell and also the direct energy source of cellular metabolism. In tumor cells, it is usually at a high level. Its intracellular concentration is above 10 mM, nearly thousands of times higher than that in the extracellular environment (<5 μM) [128,129,130], making it a promising stimulus for constructing an intelligent nanocarrier. The phenylboronic acid (PBA) group and its derivatives can selectively form ester bonds with diol compounds in aqueous solutions, and this process was reversed under conditions of lower pH and a lower concentration of diol compounds. As ATP contains diol moieties in its ribose ring, it can interact with PBA to form ester bonds. Therefore, a PBA-modified nanocarrier has been widely used in ATP-responsive gene delivery. In an example, two block catiomers were crosslinked by phenylboronate ester formed between their PBA moieties and polyol (D-gluconamide (GlcAm)) moieties, and complexed with the plasmid to form the nanoparticle. In the intracellular compartment, the high levels of ATP competitively bound with PBA and replaced GlcAm, leading to the dissociation of nanoparticles and the acceleration of plasmid release [131]. This ATP-responsive nanoparticle exhibited higher transfection efficiency in HuH-7 cells than its non-sensitive counterpart, laying the foundation for the application of PBA-modulated nanoparticles in the ATP-responsive gene delivery field. Similarly, PBA-grafted PEI has been reported to be an efficient carrier for targeted gene silencing [132]. The borate ester bond of PBA-dopamine in this carrier can be replaced by the intracellular ATP, triggering the siRNA release from the carrier. In A375 cell-bearing BALB/c nude mice, the TXPPBA/VEDA/siEGFR complexes demonstrated the best tumor inhibiting efficacy and EGFR protein knockdown among these formulations. Further, in another study, PBA-modified PEI was crosslinked with alginate via the PBA group and diol group to improve the siRNA loading ability [133]. After treatment with 5 mM ATP, both PEI-PBA and CrossPPA groups released almost all of their siRNA within 5 min, while nearly no release of siRNA was observed in 25k PEI/siRNA or 1.8k PEI/siRNA groups. The mechanism revealed that ATP-mediated CrossPPA decrosslinking and charge reversal were essential for RNA release. Apart from these mentioned polymers, this PBA-grafting strategy still works in DNA nanostructures [134].

### 4.5. External Stimulus-Responsive Nanocarriers

Besides the above nanocarriers that take advantage of the difference between tumor tissue and normal tissue, some external stimulus-responsive nanocarriers have drawn attention recently. Compared with the internal stimulus-controlled drug delivery, whose sensitivity is highly dependent on the difference between targeted and untargeted parts, external stimulus-responsive nanocarriers are less dependent on the internal signal and mainly respond to the external stimulus, such as light, the magnetic field and ultrasound. In this section, we will give some examples of external stimulus-responsive nanocarriers.

### 4.6. Magnetic Field-Controlled Gene Delivery System

Iron oxide nanoparticles have been widely used in the drug delivery field due to their unique magnetic properties, controllable size and ease of scalable production. Extensive studies have employed surface-engineered iron oxide nanoparticles for magnetic-guided drug delivery, as the iron oxide nanoparticles could actively accumulate at the desired site under the magnetic fields [135,136,137,138,139]. In a study, two differently shaped magnetic mesoporous silica nanoparticles (M-MSNs) were developed for magnetically mediated suicide gene therapy [138]. The herpes simplex virus thymidine kinase/ganciclovir (HSV-TK/GCV) gene was adsorbed onto M-MSNs, whose surface was modified by PEG-g-PLL to offer a positive charge for plasmid loading. Due to the unique magnetofection properties of these two M-MSNs, the TK gene expression was enhanced under the magnetic field. Moreover, biodistribution study revealed that the tumor accumulation of both M-MSNs was significantly increased by the magnetic field, suggesting their potential of magnetically guided tumor targeting. The Rod-like M-MSNs exhibited better tumor targeting properties than sphere-like M-MSNs regardless of the magnetic field, highlighting the critical role of shape in the preferential tumor accumulation of M-MSNs. Consistent with the tumor targeting capability, the superior antitumor efficacy was also seen in the Rod-like M-MSNs under the magnetic field in nude mice bearing HepG2 tumor xenografts. Additionally, these two M-MSNs could serve as MRI contrast agent to monitor therapeutic efficacy by MRI. Another magnetic field-guided tumor-targeted multimodal nanoplatform has been reported by Wang et al. recently [140]. They encapsulated doxorubicin (DOX) into the exosomes by electroporation, and further modified the exosomes with polydopamine (PDA) coated magnetic Fe_3_O_4_ nanoparticles (Fe_3_O_4_@PDA). A robust fluorescence signal was detected at the tumor site in mice treated by Exo-DOX-Fe_3_O_4_@PDA-MB with magnetic field while nearly no fluorescence signal was observed in mice without magnetic field, verifying the excellent magnetically targeted properties of this platform. Furthermore, the platform could achieve cooperative gene/chemo/photothermal cancer therapy by virtue of the potent photothermal properties of PDA, the cytotoxicity of DOX and miR-21-responsive molecular imaging. However, use of permanent magnetic field in this study may interfere with the interaction of magnetic nanoparticles and induce the aggregation of magnetic nanoparticles. The author reasoned that use of low-frequency pulsed magnetic fields or alternating magnetic fields may be a reasonable solution to reduce the aggregation of magnetic nanoparticles.

Inspired by the movement of natural motile bacteria E. Coli, various magnetic micro-/nanoswimmers of helical shapes have been developed such as artificial bacterial flagella (ABFs) [141]. ABFs can move under the control of low-strength rotating magnetic fields. In a study, ABFs were functionalized with lipoplex, which was formed by lipofectamine and plasmid, for targeted gene delivery in human embryonic kidney (HEK 293) cells. The functionalized ABFs were guided by the magnetic field and targeted to the HEK 293 cells for gene expression.

### 4.7. Light-Responsive Gene Delivery System

Many nanoparticles including organic or inorganic nanoparticles possess extraordinary photothermal or photodynamic characteristics [100,142,143,144], making light an attractive stimulus to control the gene delivery. Light of various wavelengths has been used in light-controlled nanocarriers, including ultraviolet light [145], visible light [146,147] and near-infrared (NIR) light [148]. Taking the coumarin-anchored polyamidoamine (PAMAM) dendrimer as an example, it can self-assemble in aqueous solution via hydrophobic interactions because of its unique amphiphilic properties and improve the binding affinity with nucleic acids because of the hydrophobic substitute coumarin. More importantly, coumarin can cyclodimerize when exposed to light of a wavelength above 300 nm while the crosslinked coumarin can be degraded into monomers under a shorter wavelength UV light. Therefore, this nanocarrier can control the drug release by UV light and minimize the side effect on normal tissues [145]. However, the limited penetrating ability of UV light may hinder their clinical application [149]. Therefore, in another example, Wang et al. developed a far-red light-mediated nanocarrier which could be triggered by far-red light (661 nm) at low optical power density [150]. Under irradiation, the incorporated photosensitizer produced non-cytotoxic levels of ROS, which could enhance endosomal escape and promote p53 gene release via degradation of thioketal-crosslinked polyethylenimine (TK-PEI). The light-responsive characteristics of this nanocarrier were verified both in vitro and in vivo. Similarly, a photolabile spherical nucleic acid (PSNA) has also been reported that was self-assembled by siRNA conjugated peptide nucleic acid-based ASO (pASO) via a ROS-sensitive linker [151]. The NIR light-triggered ROS can break the ^1^O_2_-cleavable linker, leading to the disassembly of PSNA and in turn facilitate the siRNA release. The released siRNA and pASO efficiently inhibited the expression of HIF-1α and Bcl-2 under irradiation and potently inhibit tumor growth. Other photo-responsive units such as the photosensitive platinum(IV) complexes (Pt(IV)) [148] has also been applied in constructing light-controllable gene release.

Photo-chemical internalization (PCI), a unique mechanism to escape from the endosome by destroying the endosome membranes via the produced ROS under irradiation, have shown great promise in recent years, especially in the gene delivery field [147,152,153,154]. Even plasmid of a much larger size (e.g., CRISPR-based plasmid, ≈10,000 bp) has been successfully delivered via PCI [155]. Generally, to make use of the PCI effect, the nanocarrier contains a sort of photosensitizer that can produce ROS, and this process relies on both light and oxygen. Nevertheless, the efficiency of ROS generation by photosensitizers is relatively limited due to the hypoxic tumor microenvironment. To overcome this dilemma, Zhang et al. developed a ROS-independent photoactivatable nanoparticle named CNP_PtCP/si(c-fos)_ [147]. Rather than incorporating a photosensitizer for ROS production, they synthesized a polymer containing the photoactivatable Pt(IV) prodrugs that can generate azidyl radicals (N3^•^) to facilitate N3^•^-mediated PCI. Compared with the ROS, the N3^•^ possess milder oxidation energy and are less dependent on oxygen [156], thus minimizing the damage to the co-loaded drugs and exhibiting application potential in wider field.

In addition to facilitating drug release or promoting PCI via light, strategies to modulate the gene expression under the control of light has also been applied. By incorporating a light-sensitive transcription factor (GAVPO) and a specific promoter, researchers developed a light switchable transactivator [157,158], which have the capability to initiate the gene transcription under irradiation. The PEI-based targeted nanoparticles can actively target the tumor tissues and overcome the intracellular delivery barriers, and the released plasmid can encode diphtheria toxin A to induce cell apoptosis under the irradiation of blue light [159]. Likewise, by replacing the original promoter with a heat-inducible promoter, Chen et al. devised an optogenetically activatable CRISPR-Cas9 nanosystem [64]. This nanosystem consisted of cationic polymer-coated Au nanorod and a transformed CRISPR plasmid driven by a heat-inducible promoter. Taking advantage of the excellent photothermal properties of Au nanorod, this nanosystem transformed the light into heat to trigger the expression of CRISPR plasmid, achieving spatiotemporally controllable gene editing and reducing the off-target effect. On the basis of the potent performance of this nanosystem, in their subsequent study, this nanosystem was used for cancer immunotherapy by combination of photothermal therapy-triggered immunogenic cell death (ICD) and CRISPR-mediated PD-L1 knockout for immune checkpoint blockade [66].

### 4.8. Ultrasound-Targeted Gene Delivery System

Ultrasound-targeted gene delivery (UTGD), also termed as ultrasound-targeted microbubble destruction or ultrasound mediated gene delivery, allows specifically directing the gene to the target site via ultrasound-focused techniques. Moreover, UTGD can facilitate cellular uptake by enhancing the membrane permeability when exposed to ultrasound [160,161]. Up to now, varieties of genes have been successfully delivered by UTGD. Basically, genes or nucleic acids are loaded onto ultrasonic nanoparticles via two ways: (1) direct conjugation onto the surface or (2) complexed with the cationic polymers first and then modified onto the nanoparticle [162]. Use of cationic lipids such as DPPC, DSPC and DOTAP or ionizable lipids to condense nucleic acids has been reported for UTGD. For example, Un et al. reported an ultrasound (US)-responsive gene carriers for treating metastatic and relapsed melanoma, using DSTAP, DSPC and NH_2_-PEG_2000_-DSPE or mannose-modified PEG_2000_-DSPE [163]. This mannose-modified bubble lipoplexes achieved efficient DNA vaccination under US exposure and elicited potent antitumor efficacy. In another study, Tayier et al. developed a kind of novel nanobubble for targeted gene delivery under focused ultrasound [164]. The nanobubbles were extracted from Halobacterium NRC-1 (Halo) and could produce stable ultrasound contrast signals in vitro and in vivo [165]. Further modification of PEI endowed the nanobubbles with the DNA condensation ability. Compared with the conventional microbubbles whose particle size are over 1 μm, this biosynthetic nanobubble has a much smaller size of around 200 nm, making it easier to enter the target cells. Notably, this nanobubble DNA/CBNBs significantly improved the gene transfection efficiency under ultrasound as revealed by the increased number of EGFP-positive cells. Moreover, nearly threefold of fluorescent signals were observed in the DNA/CBNBs group under ultrasound in comparison with the DNA/PEI group in a mouse model, further demonstrating the potential of ultrasound-enhanced gene delivery. Ultrasound-enhanced ROS nanocarrier has been constructed to control the gene release as well [166]. This nanocarrier was decorated with IR780, a sonosensitizer that can generate ROS under ultrasound stimulation. The generated ROS can revert the charge of the ROS-sensitive polymer, causing gene release as a result of charge repulsion for ultrasound-triggered gene delivery. Despite the advancement of ultrasound in gene delivery, special attention should be paid to the safety issues of using ultrasound due to ultrasound-associated tissue heating [162]. The non-homogeneity of sound transmission may offset the delivery efficiency and the therapeutic outcomes.

### 4.9. Multi-Stimulus Responsive Gene Delivery System

Various delivery systems that have been exploited for stimuli-responsive nucleic acids delivery are summarized in Table 1. Instead of using a single stimulus to control the gene delivery, several stimuli have been applied simultaneously in fabricating multi-stimuli responsive gene delivery systems, which has gained more and more attention recently owing to their superiority in accuracy and specificity. In this part, we will briefly show some examples of multi-stimulus responsive gene delivery systems as some have been mentioned in the above sections. For instance, Gao et al. designed a pH/redox dual-responsive polyplex for co-delivering MDR1 siRNA and doxorubicin (DOX) to overcome the multidrug resistance [167]. Their designs were as follows: firstly, a pH-sensitive PEG-b-PLA-PHis was conjugated with cationic OEI by a disulfide bond to form the mPEG-*b*-PLA-PHis-ss-OEI; then this amphiphilic polymer self-assembled into micelles and incorporated the hydrophobic DOX in the inner core; lastly, the MDR1 siRNA was adsorbed onto the micelles via electrostatic interactions. These nanoparticles can accumulate in the tumor via EPR effect, and after entering the tumor cells via endocytosis, the decreased pH and abundant GSH in the endo-lysosome triggered the payload release. Both the release of DOX and siRNA from the nanoparticles was accelerated under acidic pH, and the siRNA release was further facilitated by dithiothreitol (DTT), verifying the pH and redox-responsive behavior of this nanoparticle. The potent antitumor efficacy in MCF-7/ADR tumor xenografted nude mice further demonstrated the feasibility of using pH/redox dual-responsive nanocarriers for tumor treatment. Likewise, Zhang et al. designed a dual-locking nanoparticle (DLNP) that can distinguish tumor and normal tissues in response to the pH and the H_2_O_2_ concentration in the tumor microenvironment [63]. DLNP contained a cationic core formed by the CRISPR/Cas13a plasmid and 4-(hydroxymethyl) phenylboronic acid (HPBA)-modified polyethyleneimine (PEI1.8k–HPBA), and the core was coated with cis-aconitic anhydride (CA) and sodium glucoheptonate dehydrate (SGD)-modified poly(ethylene glycol)-b-polylysine (mPEG_113_-b-PLys_120_/SGD_5_/CA). In the acid tumor microenvironment, the CA of the outlayer was decomposed, thus turning the anionic mPEG_113_-b-PLys_120_/SGD_5_/CA to the cationic mPEG_113_-b-PLys_120_/SGD_5_ and leading to the detachment of the outlayer coating. Meanwhile, HPBA was cleaved from the PEI1.8k–HPBA by the high levels of H_2_O_2_ at the tumor site. The pH-H_2_O_2_ dual responsive characteristics of this nanocarrier enabled it to restrict the gene editing by CRISPR/Cas13a in tumor tissues and significantly reduce the undesired activations of CRISPR/Cas13a in normal tissues. In another example, taking advantage of the alterations of HAase or GSH in the tumor microenvironment, a multi-responsive “turn-on” polyelectrolyte complex was fabricated to achieve highly efficient gene transfection in vitro and in vivo [123]. The results demonstrated that this multi-responsive delivery system outperformed their single-responsive counterparts in transfection efficiency both in vitro and in vivo.

## 5. Conclusions and Outlooks

Despite the rapid development of these nucleic acids-based gene therapies, efficient delivery remains one of the most challenging obstacles for the clinical use of nucleic acids. Viral vectors represent the most efficient strategies for gene delivery owing to their high transfection efficiency, but their use is still impeded by the safety issues, package limit and high cost. The emergence of a delivery system provides an alternative option to the gene delivery. Nanoparticles are capable of delivering various nucleic acids (i.e., siRNA, miRNA, mRNA, plasmid and CRISPR-based machinery) of difference size without the package limit. Moreover, due to their facile modification, nanoparticles can be endowed with multiple functions to facilitate their delivery in the face of multiple barriers. Efficient endosome escape remains a key aspect of transfection efficiency. To achieve high transfection efficiency, ideal nanoparticles should have the capacity to protect the loaded nucleic acids from damage, evade the degradation of lysosome and locate the nucleic acids at their target sites. Recent years have witnessed the great progress of nanoparticles in the gene delivery field; although promising, nanoparticles that respond to internal or external stimuli are more desired, especially in treating diseases such as tumors. The unique properties of the stimuli responsiveness of these nanoparticles make it possible to precisely guide the therapeutic nucleic acids to the target site and eliminate the potential off-target effects of the gene agents. Further, these stimuli-responsive nanoparticles provide the possibility of releasing the gene agents in an intelligent manner, significantly improving the delivery efficiency. Moreover, some of these stimuli-responsive nanoparticles can serve as a theranostical platform to simultaneously treat diseases and monitor the progression of diseases by virtue of their excellent imaging performance or the introduction of contrast agents [168,169,170]. Despite the aforementioned advantages of stimuli-responsive nanoparticles, several critical issues remain to be solved for advancing their clinical use. First of all, although tremendous efforts have been devoted to modifying the materials or synthesizing the chemical substances via combinatorial chemistry for gene delivery, the overall delivery efficiency of non-viral vehicles is inferior to that of viral vectors. With respect to the large plasmids, such as CRISPR-based machinery, the performance of nanoparticles suffers a major setback. Sometimes, the most common method using the optimal nanoparticles screened in vitro for in vivo use may not work, as the delivery efficiency in vivo may not coincide with that in vitro. Secondly, the accumulation of nanoparticles at the target site remains low, despite the use of various modification strategies, such as the decoration of ligands, antibodies, aptamers or even cell membranes to enhance the targetability. Only a very few of the administrated nanoparticles can reach the tumor, not to mention their accumulation in the target cells [171]. The tendency of nanoparticles to preferably accumulate at the liver makes it challenging to target other diseased tissues [172,173] and raises concerns about the potential damage to the liver, which relates to the off-target effects of the nanoparticles. Moreover, fabrication of stimuli-responsive nanoparticles is much harder owing to their sophisticated structure, but considering the dosages used in animal models, their production at a small scale can meet the demands of preclinical use. However, their clinical use may be challenged by the difficulties of their massive production. Moreover, the toxicity of nanoparticles themselves should also be brought to attention for their clinical use, especially for inorganic nanoparticles and polymers that are recognized as hard to degrade. Most studies only focus on the safety of nanoparticles during the therapeutic period instead of their long-term safety by using several simplified parameters as the safety indicators, such as body weight, organ index, blood routine and histological observation. Additionally, despite the similarities in some pathological changes of tumors, the internal signals may vary from one tumor model to another model or even from person to person, making it even more challenging to develop a universal delivery platform suitable for various tumor models. Use of external signals that are independent of these pathological changes of tumors may be a better choice in this regard. It is worth mentioning that although multiple-stimuli-responsive nanocarriers seem more promising, as they are capable of responding to multiple stimuli, whether these more complex systems provide sufficiently improved delivery parameters to justify the extra effort involved remains to be seen [29].

Therefore, we still have a long way to go for the advancement of nucleic acid-based therapies. As mentioned above, future efforts should be made in the following aspects: (1) having a better understanding of the structure–activity relationship of materials and the interactions of nucleic acids with nanoparticles to guide the design of nanoparticles with a high delivery efficiency; (2) use of more reliable models for evaluating gene delivery efficiency instead of the monolayer cells; (3) improving tumor accumulation by optimizing the parameters of nanoparticles or some innovative methods to minimize the off-target effects on normal tissues; (4) keeping a balance of the functional diversity and degree of production difficulty of the stimuli-responsive nanoparticles during design, and using technologies such as microfluidic mixing or developing methods for producing nanoparticles at a large scale to facilitate their clinical translation; (5) elaborating the in vivo processes of nanoparticles thoroughly and paying attention to the long-term safety of nanoparticles, including their degradability. With the rapid development of nanomedicine and the in-depth knowledge of diseases, we hope that more efficient and more intelligent stimuli-responsive nanoparticles show their value in the clinic in the near future.

## Figures and Tables

**Figure 1 pharmaceutics-15-01450-f001:**
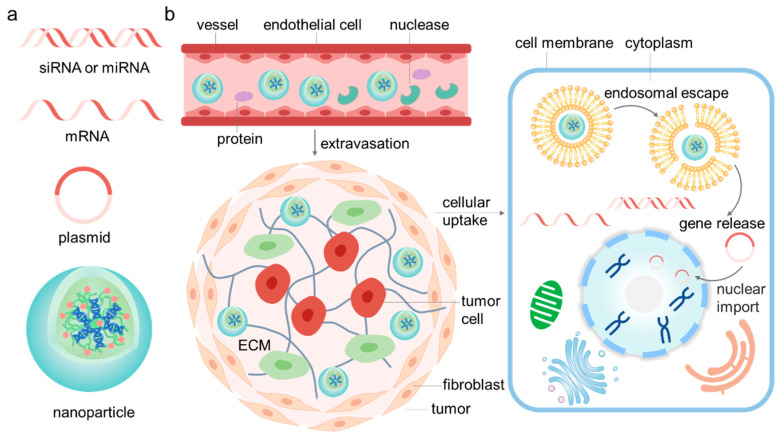
Schematic illustration of the multiple barriers for gene delivery in vivo. (**a**) Firstly, the therapeutic nucleic acids (e.g., siRNA, mRNA or plasmid) must be condensed into uniform small nanoparticles (preferably less than 100 nm). (**b**) During systematic circulation, the nanoparticles must remain stable, evade clearance by MPS and protect the nucleic acids from the degradation of nuclease in the complicated physiological environment. Moreover, they should be capable of extravasating from the vessels and target the diseased site, such as a tumor. To reach their target cells (e.g., tumor cells), nanoparticles have to cross the barriers formed by the various cells and the extracellular matrixes (ECMs) in the tumor microenvironment. Moreover, efficient cellular uptake is critical for gene delivery. Most nanoparticles enter the cells via endocytosis and are entrapped in the endosome. Nanoparticles must escape from the endosome and unload the encapsulated nucleic acids to fulfill their functions. Additionally, for plasmid, the existence of nuclear membranes represents another intracellular barrier for nuclear import.

**Figure 2 pharmaceutics-15-01450-f002:**
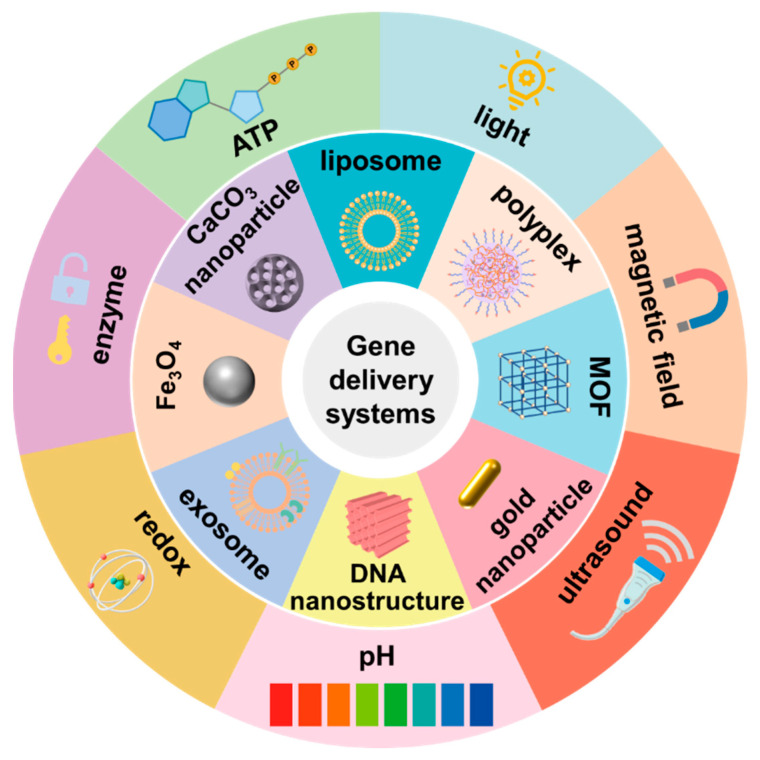
Overview of the main stimuli-responsive nanoparticles in gene therapy.

**Figure 3 pharmaceutics-15-01450-f003:**
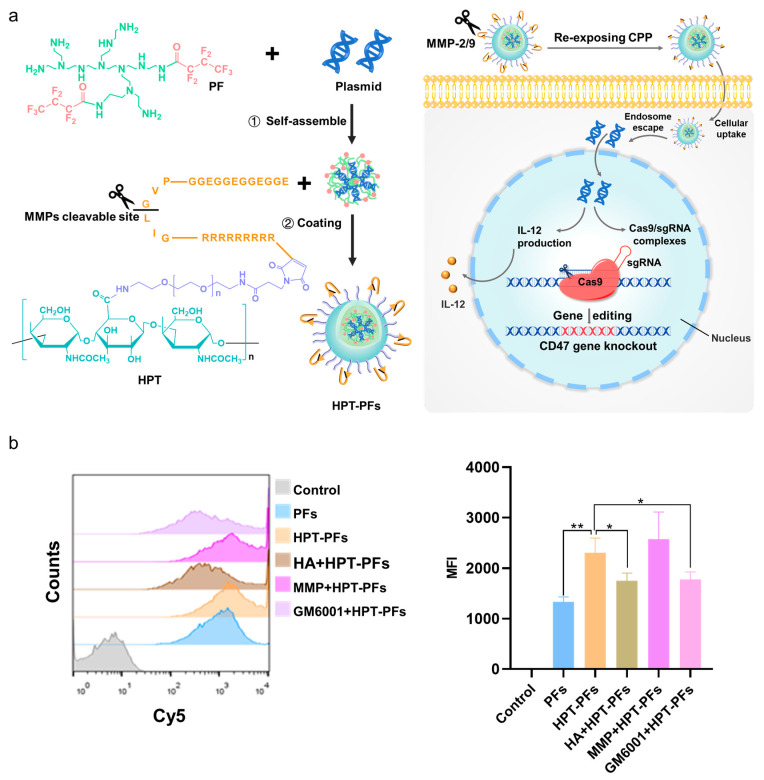
(**a**) Preparation of HPT-PFs via self-assembly of two modules. First, plasmids for CD47 knockout or IL-12 production were condensed by PF module (fluorinated polyethylenimine) via electrostatic interactions to form the core. Then, the core was further coated by HPT module (hyaluronic acid-polyethylene glycol-tumor microenvironment-sensitive peptides, HA-PEG-TMSPs) to assemble into HA and TMSP dual-modified nanoparticles. At the tumor circumstances, the high expression of MMPs cleave the TMSP and exposure of cellular penetrating peptides (CPPs), which promotes the endocytosis of nanoparticles into the tumor cells. In cells, the nanoparticle could rapidly escape from endosome by “proton sponge effect” and entered the nuclei. Plasmids fulfill their functions, including IL-12 production and Cas9/sgRNA complexes expression. The Cas9/sgRNA complexes can specifically cleave the target site at CD47 gene under the guidance of sgRNA and knock out CD47. (**b**) Cellular uptake of HPT-PFs in B16F10 cells under various conditions determined by flow cytometry and corresponding statistical fluorescence intensity. HA + HPT − PFs, MMP + HPT − PFs and GM6001 + HPT − PFs: pretreatment with HA (2 mg/mL), activated collagenase IV (2 μg/mL) and a MMP inhibitor GM6001 (2.5 μg/mL) for 2 h, respectively, followed by transfection (n = 3). * *p* < 0.05, ** *p* < 0.01. Adapted with permission from [116]. Copyright 2022, Elsevier.

**Table 1 pharmaceutics-15-01450-t001:** Summary of stimuli-responsive delivery systems.

Stimuli	Delivery Systems	Sensitive Linkers/Materials	Drugs	References
**pH**	PEI	Schiff base bond	Luciferase plasmid	[27]
	Peptide nanoparticles (PNPs)	Peptide	Tissue factor (TF) siRNA	[28]
	Metal–organic framework (MOF) nanoparticle ZIF-8	ZIF-8	miR-34a mimic	[69]
	Solid lipid nanoparticles	Imine bond	miR-200 and irinotecan	[74]
	PEI−PBA/miR146a/PEI−DMA-C225, PMPC polyplexes	Ester bond	miR146a	[75]
	CaP-phospholipid complexes	CaP	As_2_O_3_ and HER2 siRNA	[77]
	Calcium carbonate nanohybrids	Calcium carbonate	p53 plasmid	[78]
	Dextran-quantum dot nanohybrids	Schiff base bond	p53 plasmid	[79]
	Black phosphorus nanosheets	Black phosphorus nanosheets	HTERT siRNA	[80]
	Lipid nanoparticles (LNP)	Ionizable lipids	CD47 siRNA and PD-L1 siRNA	[85]
**ROS**	Heparin nanoparticles	Disulfide bond	miR-499	[82]
	Substance P (SP)-crosslinked BPEI	Boric acid ester bond	Plk1 siRNA	[89]
	DNA nanostructures	Disulfide bond	Doxorubicin, Bcl2 siRNA and P-gp siRNA	[93]
	Arginine-modified poly(disulfide amine)/siRNA nanocomplexes	Disulfide bond	KRAS siRNA	[95]
	Heparin nanoparticles	Disulfide bond	CRISPR/Cas 9 plasmid targeting survivin	[96]
	Fusogenic lipidic polyplexes	Boric acid ester bond	TRAIL plasmid	[97]
	Poly[(2-acryloyl)ethyl(p-boronic acid benzyl) diethylammonium Bromide] (B-PDEAEA)	Boric acid ester bond	TRAIL plasmid	[98]
	Polyphotosensitizers (pPSs)	Thioketal	HIF-1α siRNA	[100]
**Enzyme**	Cationic polymer PQDEA	Acetyloxybenzyl ester	TRAIL plasmid	[102]
	Liposome-based layer-by-layer nanoparticles	MMP9-sensitive peptide	Luciferase plasmid	[107]
	Amphiphilic Dendrimer engineered nanocarrier system (ADENS)	MMP2/9-sensitive peptide	Paclitaxel and VEGF siRNA	[114]
	PEI-based nanoparticles HPT-PF	MMP2/9-sensitive peptide	IL-12 plasmid and CRISPR/Cas 9 plasmid targeting CD47	[115]
	Polyelectrolyte complexes	HA	Luciferase plasmid	[123]
	Hyaluronic acid (HA) coated nanoassembly	HA	Bcl-2 shRNA encoded plasmid and doxorubicin	[124]
	Polymetformin (pmet)-based nanosystem	HA	IL-12 plasmid	[125]
	Peptide	HA	LOX-1 siRNA	[126]
**ATP**	Polyplex Micelles	Phenylboronic acid (PBA) group	Luciferase plasmid	[131]
	4-carboxyphenylboronic acid (PBA) and dopamine Grafted vitamin E (VEDA)	Phenylboronic acid (PBA) group	Eg5 siRNA and EGFR siRNA	[132]
	PEI-PBA	Phenylboronic acid (PBA) group	Bcl2 siRNA	[133]
	DNA nanostructures	ATP aptamer	Plk1 siRNA	[134]
**Magnetic field**	Magnetic mesoporous silica nanoparticles (M-MSNs)	Iron oxide nanoparticles	The herpes simplex virus thymidine kinase/ganciclovir (HSV-TK/GCV) plasmid	[138]
	Exosome-based platforms	Iron oxide nanoparticles	DOX and molecular beacon targeting the miR-21	[140]
	Lipoplexes functionalized artificial bacterial flagella	Artificial bacterial flagella	pDNA encoding yellow–green fluorescent Venus protein	[141]
**Light**	Gold Nanorod	HSP70 promoter	CRISPR/Cas 9 plasmid targeting AAVS1	[64]
	Gold Nanorod	HSP70 promoter	CRISPR/Cas 9 plasmid targeting PD-L1	[66]
	Coumarin modified PAMAM	Coumarin	TRAIL plasmid	[145]
	Prodrug-backboned polymeric nanoparticle system	Pt(IV) prodrug	Pt(IV) prodrug and c-fos siRNA	[147]
	TK-PEI/HAP/p53 NCs	Pheophytin a	p53 plasmid	[150]
	Spherical nucleic acid	Pheophorbide a	HIF-1α siRNA and Bcl2 siRNA	[151]
**Ultrasound**	Mannose-modified bubble lipoplexes	Bubble lipoplexes	Plasmid co-expressing ubiquitylated gp100 and TRP-2	[163]
	Cationic biosynthetic Nanobubble (CBNB)	Nanobubble	The pEGFP and pCMV-Luc reporter plasmids	[164]
	B-PDEAEA loaded liposome	IR780	TRAIL plasmid	[166]

## Data Availability

Not applicable.

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
