# Peer review of "Advances and Challenges of Stimuli-Responsive Nucleic Acids Delivery System in Gene Therapy"

_pharmaceutics, 2023, doi:10.3390/pharmaceutics15051450_

Round 1

Reviewer 1 Report

General comments:

This manuscript provides an excellent review of the various delivery systems used for nucleic acid-based gene therapy.

This reviewer suggests that a copy editor should evaluate the manuscript to correct errors in grammar, punctuation, and spelling.

Specific comments:

Abstract, line 14-15:  Formatting issue in these lines.

Page 5, line 7:  Insert “been” after “has”.

Page 5, line 16:  “complicate” should be “complicated”.

Page 6, line 3:  “disassemble) should be “disassembly”.

Page 10, line 2:  “resulted” should be “result”.

Page 11, lines 4-5:  “in varieties of tumors” should be “in a variety of tumors”.

Page 11, lines 20-21 from bottom:  “system has been fabricated” should be “systems have been fabricated”.

Page 15, line 5:  “participant” should be “participate”.

Page 15, line 14:  “a novel hyaluronidases-triggered” should be “a novel hyaluronidase-triggered”.

Page 15, line 27:  “energy sources” should be “energy source”.

Page 16, line 7:  “drew” should be “drawn”.

Page 16, line 14:  “nanoparticles has been” should be “nanoparticles have been”.

Page 17, line 6 from bottom:  “which have” should be “which has”.

This reviewer suggests that a copy editor should evaluate the manuscript to correct errors in grammar, punctuation, and spelling.

Author Response

Thank you very much for your affirmation and professional suggestions to our article. We have checked the manuscript and corrected the typos. The manuscript has been checked by a colleague fluent in English writing.

Reviewer 2 Report

The paper is well written and full of interesting ideas. Since it is a review, it is suggested to broaden the bibliography, in particular by looking at works on similar topics (reviews, in particular) that have appeared in recent times, such as for example

Wirth, T., Parker, N., & Ylä-Herttuala, S. (2013). History of gene therapy. Gene, 525(2), 162-169.

Xu, H., Li, Z., & Si, J. (2014). Nanocarriers in gene therapy: a review. Journal of biomedical nanotechnology, 10(12), 3483-3507.

Bulcha, J. T., Wang, Y., Ma, H., Tai, P. W., & Gao, G. (2021). Viral vector platforms within the gene therapy landscape. Signal transduction and targeted therapy, 6(1), 53.

Wang, D., Tai, P. W., & Gao, G. (2019). Adeno-associated virus vector as a platform for gene therapy delivery. Nature reviews Drug discovery, 18(5), 358-378.

Mi, P. (2020). Stimuli-responsive nanocarriers for drug delivery, tumor imaging, therapy and theranostics. Theranostics, 10(10), 4557.

Zhao, W., Zhao, Y., Wang, Q., Liu, T., Sun, J., & Zhang, R. (2019). Remote light‐responsive nanocarriers for controlled drug delivery: Advances and perspectives. Small, 15(45), 1903060.

Author Response

 Thanks for your advice. We have read the reviews which are closely related to our review, and the reviews about history of gene therapy has been cited as [1] on page 1 in the revised manuscript. As similar reviews on viral vector platforms and stimuli-responsive nanocarriers have been included in our manuscript, these reviews are not included in. 

Reviewer 3 Report

This work by Meng Lin and Xianrong Qi is a review of different kind of DNA molecules delivered by stimuli gene delivery systems which includes their characteristics, advantages, and disadvantages. Specifically, the authors comment on pH-responsive gene release, ROS-responsive nanocarriers, enzyme-responsive nanocarriers, ATP-responsive nanocarriers, and external stimulus such as magnetic field-controlled, light-responsive, and ultrasound-targeted gene delivery systems. They report 171 references.

It is an extenseive review that may be useful for readers of Pharmaceutics interested in gene-delivery

I have one suggestion to improve the manuscript:

1) To mention in the Introduction section  two other types of DNA molecules that are used in gene therapy besides siRNA, miRNA, plasmids or CRISPR. 

One of them is Antisense oligonucleotides (ASO) that are also used extensively in gene therapy. I would suggest to include a review reference on that:

Hideo Takakusa, Norihiko Iwazaki, Makiya Nishikawa, Tokuyuki Yoshida, Satoshi Obika, and Takao Inoue. Drug Metabolism and Pharmacokinetics of Antisense Oligonucleotide Therapeutics: Typical Profiles, Evaluation Approaches, and Points to Consider Compared with Small Molecule Drugs.
Nucleic Acid Therapeutics.Apr 2023.83-94.http://doi.org/10.1089/nat.2022.0054

And the other one is the PolyPurine Reverse Hoogsteen (PPRH) hairpins which are an emergent kind of therapeutic oligonucleotides. In this case, another reference to a review could be used such as: 

Véronique Noé, Eva Aubets, Alex J. Félix, Carlos J. Ciudad. Nucleic acids therapeutics using PolyPurine Reverse Hoogsteen hairpins. Biochemical Pharmacology,Volume 189, 2021, https://doi.org/10.1016/j.bcp.2020.114371.

Minor typos or expressions were found; e.g.

Page 2: "duo" should be "due"

Page 6: "enormous" nanocarriers would read better as "vast amount" of nanocarriers, or extensive amount.

Author Response

 Thanks for your suggestions. We have added two other types of DNA molecules——ASO and PPRH on page 1. And the review references have also been included. Besides, the typos you mentioned have been corrected.